# Multi-View Reinforcement Learning

**Minne Li** *
University College London
London, United Kingdom
minne.li@cs.ucl.ac.uk

**Lisheng Wu** *
University College London
London, United Kingdom
lisheng.wu.17@ucl.ac.uk

**Haitham Bou Ammar** †
University College London
London, United Kingdom
haitham.bouammar71@googlemail.com

**Jun Wang**
University College London
London, United Kingdom
junwang@cs.ucl.ac.uk

## Abstract

This paper is concerned with multi-view reinforcement learning (MVRL), which allows for decision making when agents share common dynamics but adhere to different observation models. We define the MVRL framework by extending partially observable Markov decision processes (POMDPs) to support more than one observation model and propose two solution methods through observation augmentation and cross-view policy transfer. We empirically evaluate our method and demonstrate its effectiveness in a variety of environments. Specifically, we show reductions in sample complexities and computational time for acquiring policies that handle multi-view environments.

## 1 Introduction

In reinforcement learning (RL), tasks are defined as Markov decision processes (MDPs) with state and action spaces, transition models, and reward functions. The dynamics of an RL agent commence by executing an action in a state of the environment according to some policy. Based on the action choice, the environment responds by transitioning the agent to a new state and providing an instantaneous reward quantifying the quality of the executed action. This process repeats until a terminal condition is met. The goal of the agent is to learn an optimal action-selection rule that maximizes total-expected returns from any initial state. Though minimally supervised, this framework has become a profound tool for decision making under uncertainty, with applications ranging from computer games [24, 30] to neural architecture search [42], robotics [3, 25, 27], and multi-agent systems [22, 34, 38, 40].

Common RL algorithms, however, only consider observations from one view of the state space [17]. Such an assumption can become too restrictive in real-life scenarios. To illustrate, imagine designing an autonomous vehicle that is equipped with multiple sensors. For such an agent to execute safe actions, data-fusion is necessary so as to account for all available information about the world. Consequently, agent policies have now to be conditioned on varying state descriptions, which in turn, lead to challenging representation and learning questions. In fact, acquiring good-enough policies in the multi-view setting is more complex when compared to standard RL due to the increase in sample complexities needed to reason about varying views. If solved, however, multi-view RL will allow for data-fusion, fault-tolerance to sensor deterioration, and policy generalization across domains.

Numerous algorithms for multi-view learning in supervised tasks have been proposed. Interested readers are referred to the survey in [23, 39, 41], and references therein for a detailed exposition.

Though abundant in supervised learning tasks, multi-view data fusion for decision making has gained less attention. In fact, our search revealed only a few papers attempting to target this exact problem. A notable algorithm is the work in [6] that proposed a double task deep Q-network for multi-view reinforcement learning. We believe the attempt made by the authors handle the multi-view decision problem indirectly by carrying innovations from computer-vision, where they augment different angle cameras in one state and feed to a standard deep Q-network. Our attempt, on the other hand, aims to resolve multi-view decision making directly by learning joint models for autonomous planning. As a by-product of our method, we arrive at a learning pipeline that allows for improvements in both policy learning and feature representations.

Closest to our work are algorithms from the domain of partially observable Markov decision processes (POMDPs) [17]. There, the environment's state is also hidden, and the agent is equipped with a sensor (i.e., an observation function) for it to build a belief of the latent state in order to execute and learn its policy. Although most algorithms consider one observation function (i.e., one view), one can generalize the definition of POMDPs, to multiple types of observations by allowing for a joint observation space, and consequently a joint observation function, across varying views. Though possible in principle, we are not aware of any algorithm from the POMDP literature targeting this scenario. Our problem, in fact, can become substantially harder from both the representation and learning perspectives. To illustrate, consider a POMDP with only two views, the first being images, while the second a low-dimensional time series corresponding to, say joint angles and angular velocities. Following the idea of constructing a joint observation space, one would be looking for a map from history of observations (and potentially actions) to a new observation at the consequent time steps. Such an observation can be an image, a time series, or both. In this setting, constructing a joint observation mapping is difficult due to the varying nature of the outputs and their occurrences in history. Due to the large sample and computational complexities involved in designing larger deep learners with varying output units, and switching mechanisms to differentiate between views, we rather advocate for a more grounded framework by drawing inspiration from the well-established multi-view supervised learning literature. Thus, leading us to multi-view reinforcement learning[3].

Our framework for multi-view reinforcement learning also shares similarities with multi-task reinforcement learning, a framework that has gained considerable attention [3, 10, 11, 12, 26, 32, 36]. Particularly, one can imagine multi-view RL as a case of multi-task RL where tasks share action spaces, transition models, and reward functions, but differ in their state-representations. Though a bridge between multi-view and multi-task can be constructed, it is worth mentioning that most works on multi-task RL consider same observation and action spaces but varying dynamics and/or reward functions [33]. As such, these methods fail to handle fusion and generalization across feature representations that vary between domains. A notable exception is the work in [2], which transfers knowledge between task groups, with varying state and/or action spaces. Though successful, this method assumes model-free with linear policy settings. As such, it fails to efficiently handle high-dimensional environments, which require deep network policies.

In this paper, we contribute by introducing a framework for multi-view reinforcement learning that generalizes partially observable Markov decision processes (POMDPs) to ones that exhibit multiple observation models. We first derive a straight-forward solution based on state augmentation that demonstrates superior performance on various benchmarks when compared to state-of-the-art proximal policy optimization (PPO) in multi-view scenarios. We then provide an algorithm for multi-view model learning, and propose a solution capable of transferring policies learned from one view to another. This, in turn, greatly reduces the amount of training samples needed by around two order of magnitudes in most tasks when compared to PPO. Finally, in another set of experiments, we show that our algorithm outperforms PILCO [9] in terms of sample complexities, especially on high-dimensional and non-smooth systems, e.g., Hopper[4]. Our contributions can be summarized as:

- formalizing multi-view reinforcement learning as a generalization of POMDPs;
- proposing two solutions based on state augmentation and policy transfer to multi-view RL;
- demonstrating improvement in policy against state-of-the-art methods on a variety of control benchmarks.

## 2  Multi-View Reinforcement Learning

This section introduces multi-view reinforcement learning by extending MDPs to allow for multiple state representations and observation densities. We show that POMDPs can be seen as a special case of our framework, where inference about latent state only uses *one* view of the state-space.

### 2.1  Multi-View Markov Decision Processes

To allow agents to reason about varying state representations, we generalize the notion of an MDP to a multi-view MDP, which is defined by the tuple $\mathcal{M}_{\text{multi-view}} = \left\langle \mathcal{S}, \mathcal{A}, \mathcal{P}, \mathcal{O}_1, \mathcal{P}_{\text{obs}}^1, \ldots, \mathcal{O}_N, \mathcal{P}_{\text{obs}}^N \right\rangle$. Here, $\mathcal{S} \subseteq \mathbb{R}^d$ and $\mathcal{A} \subseteq \mathbb{R}^m$, and $\mathcal{P} : \mathcal{S} \times \mathcal{A} \times \mathcal{S} \to [0,1]$ represent the standard MDP's state and action spaces, as well as the transition model, respectively.

Contrary to MDPs, multi-view MDPs incorporate additional components responsible for formally representing multiple views belonging to different observation spaces. We use $\mathcal{O}_j$ and $\mathcal{P}_{\text{obs}}^j : \mathcal{S} \times \mathcal{O}_j \to [0,1]$ to denote the observation space and observation-model of each sensor $j \in \{1, \ldots, N\}$. At some time step $t$, the agent executes an action $\boldsymbol{a}_t \in \mathcal{A}$ according to its policy that conditions on a history of *heterogeneous* observations (and potentially actions) $\mathcal{H}_t = \{\boldsymbol{o}_1^{i_1}, \ldots, \boldsymbol{o}_t^{i_t}\}$, where $\boldsymbol{o}_k^{i_k} \sim \mathcal{P}_{\text{obs}}^{i_k} (.|\boldsymbol{s}_k)$ for $k \in \{1, \ldots, T\}$ and[5] $i_k \in \{1, \ldots, N\}$. We define $\mathcal{H}_t$ to represent the history of observations, and introduce a superscript $i_t$ to denote the type of view at the $t^{th}$ time instance. Each $i_t$ is dependent on the observation function (availability of specific views) and conditioned on the environment's state. As per our definition, we allow for $N$ different types of observations, therefore, $i_t$ is allowed to vary from one to $N$. Depending on the selected action, the environment then transitions to a successor state $\boldsymbol{s}_{t+1} \sim \mathcal{P}(.|\boldsymbol{s}_t, \boldsymbol{a}_t)$, which is not directly observable by the agent. On the contrary, the agent only receives a successor view $\boldsymbol{o}_{t+1}^{i_{t+1}} \sim \mathcal{P}_{\text{obs}}^{i_{t+1}} (.|\boldsymbol{s}_{t+1})$ with $i_{t+1} \in \{1, \ldots, N\}$.

### 2.2  Multi-View Reinforcement Learning Objective

As in standard literature, the goal is to learn a policy that maximizes total expected return. To formalize such a goal, we introduce a multi-view trajectory $\boldsymbol{\tau}^{\mathcal{M}}$, which augments standard POMDP trajectories with multi-view observations, i.e., $\boldsymbol{\tau}^{\mathcal{M}} = [\boldsymbol{s}_1, \boldsymbol{o}_1^{i_1}, \boldsymbol{a}_1, \ldots, \boldsymbol{s}_T, \boldsymbol{o}_T^{i_T}]$, and consider finite horizon cumulative rewards computed on the real state (hidden to the agent) and the agent's actions. With this, the goal of the agent to determine a policy to maximize the following optimization objective:

$$\max_{\pi^{\mathcal{M}}} \mathbb{E}_{\boldsymbol{\tau}^{\mathcal{M}}} \left[ \mathcal{G}_T \left( \boldsymbol{\tau}^{\mathcal{M}} \right) \right], \text{ where } \boldsymbol{\tau}^{\mathcal{M}} \sim p_{\pi^{\mathcal{M}}} \left( \boldsymbol{\tau}^{\mathcal{M}} \right) \text{ and } \mathcal{G}_T \left( \boldsymbol{\tau}^{\mathcal{M}} \right) = \sum_t \gamma^t \mathcal{R}(\boldsymbol{s}_t, \boldsymbol{a}_t), \quad (1)$$

where $\gamma$ is the discount factor. The last component needed for us to finalize our problem definition is to understand how to factor multi-view trajectory densities. Knowing that the trajectory density is that defined over joint observation, states, and actions, we write:

$$
\begin{aligned}
p_{\pi^{\mathcal{M}}} \left( \boldsymbol{\tau}^{\mathcal{M}} \right) =& \mathcal{P}_{\text{obs}}^{i_T} \left( \boldsymbol{o}_T^{i_T} | \boldsymbol{s}_T \right) \mathcal{P} \left( \boldsymbol{s}_T | \boldsymbol{s}_{T-1}, \boldsymbol{a}_{T-1} \right) \pi^{\mathcal{M}} \left( \boldsymbol{a}_{T-1} | \mathcal{H}_{T-1} \right) \ldots \mathcal{P}_{\text{obs}}^{i_1} \left( \boldsymbol{o}_1^{i_1} | \boldsymbol{s}_1 \right) \mathcal{P}_0(\boldsymbol{s}_1) \\
=& \mathcal{P}_0(\boldsymbol{s}_1) \mathcal{P}_{\text{obs}}^{i_1} \left( \boldsymbol{o}_1^{i_1} | \boldsymbol{s}_1 \right) \pi^{\mathcal{M}} \left( \boldsymbol{a}_1 | \mathcal{H}_1 \right) \prod_{t=2}^{T} \mathcal{P}_{\text{obs}}^{i_t} \left( \boldsymbol{o}_t^{i_t} | \boldsymbol{s}_t \right) \mathcal{P}(\boldsymbol{s}_t | \boldsymbol{s}_{t-1}, \boldsymbol{a}_{t-1}) \pi^{\mathcal{M}} \left( \boldsymbol{a}_{t-1} | \mathcal{H}_{t-1} \right),
\end{aligned}
$$

with $\mathcal{P}_0(\boldsymbol{s}_1)$ being the initial state distribution. The generalization above arrives with additional sample and computational burdens, rendering current solutions to POMDPs impertinent. Among various challenges, multi-view policy representations capable of handling varying sensory signals can become expensive to both learn and represent. That being said, we can still reason about such structures and follow a policy-gradient technique to learn the parameters of the network. However, a multi-view policy network needs to adhere to a crucial property, which can be regarded as a special case of representation fusion networks from multi-view representation learning [23, 39, 41]. We give our derivation of general gradient update laws following model-free policy gradients in Sect. 3.1.

Contrary to standard POMDPs, our trajectory density is generalized to support multiple state views by requiring different observation models through time. Such a generalization allows us to advocate a more efficient model-based solver that enables cross-view policy transfer (i.e., conditioning one view policies on another) and few-shot RL, as we shall introduce in Sect. 3.2.

# 3  Solution Methods

This section presents our model-free and model-based approaches to solving multi-view reinforcement learning. Given a policy network, our model-free solution derives a policy gradient theorem, which can be approximated using Monte Carlo and history-dependent baselines when updating model parameters. We then propose a model-based alternative that learns joint dynamical (and emission) models to allow for control in the latent space and cross-view policy transfer.

## 3.1  Model-Free Multi-View Reinforcement Learning through Observation Augmentation

The type of algorithm we employ for model free multi-view reinforcement learning falls in the class of policy gradient algorithms. Given existing advancements on Monte Carlo estimates, the variance reduction methods (e.g., observation-based baselines $\mathcal{B}_\phi(\mathcal{H}_t)$), and the problem definition in Eq. (1), we can proceed by giving the rule of updating the policy parameters $\boldsymbol{\omega}$ as:

$$\boldsymbol{\omega}^{k+1} = \boldsymbol{\omega}^k + \eta^k \frac{1}{M} \sum_{j=1}^{M} \sum_{t=1}^{T} \nabla_{\boldsymbol{\omega}} \log \pi^{\mathcal{M}} \left( \boldsymbol{a}_t^j | \mathcal{H}_t^j \right) \left( \mathcal{R}(\boldsymbol{s}_t^j, \boldsymbol{a}_t^j) - \mathcal{B}_\phi(\mathcal{H}_t^j) \right). \qquad (2)$$

Please refer to the appendix for detailed derivation. While a policy gradient algorithm can be implemented, the above update rule is oblivious to the availability of multiple views in the state space closely resembling standard (one-view) POMDP scenarios. This only increases the number of samples required for training, as well as the variance in the gradient estimates. We thus introduce a straight-forward model-free MVRL algorithm by leveraging fusion networks from multi-view representation learning. Specifically, we assume that corresponding observations from all views, i.e., observations that sharing the same latent state, are accessible during training. Although the parameter update rule is exactly the same as defined in Eq. (2), this method manages to utilize the knowledge of shared dynamics across different views, thus being optimal than independent model-free learners, i.e., regarding each view as a single environment and learn the policy.

## 3.2  Model-Based Multi-View Reinforcement Learning

We now propose a model-based approach that learns approximate transition models for multi-view RL allowing for policies that are simpler to learn and represent. Our learnt model can also be used for cross-view policy transfer (i.e., conditioning one view policies on another), few-shot RL, and typical model-based RL enabling policy improvements through back-propagation in learnt joint models.

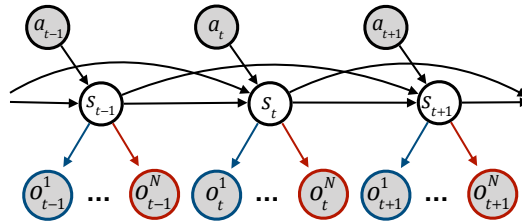

Figure 1: The graphical model of multi-view learning.

### 3.2.1  Multi-View Model Learning

The purpose of model-learning is to abstract a hidden joint model shared across varying views of the state space. For accurate predictions, we envision the generative model in Fig. (1). Here, observed random variables are denoted by $\boldsymbol{o}_t^{i_t}$ for a time-step $t$ and $i_t \in \{1, \dots, N\}$, while $\boldsymbol{s}_t \in \mathbb{R}^d$ represents latent variables that temporally evolve conditioned on applied actions. As multiple observations are allowed, our model generalizes to multi-view by supporting varying emission models depending on the nature of observations. Crucially, we do not assume Markov transitions in the latent space as we believe that reasoning about multiple views requires more than one-step history information.

To define our optimization objective, we follow standard variational inference. Before deriving the variational lower bound, however, we first introduce additional notation to ease exposure. Recall that $\boldsymbol{o}_t^{i_t}$ is the observation vector at time-step $t$ for view $i_t$[6]. To account for action conditioning, we further augment $\boldsymbol{o}_t^{i_t}$ with executed controls, leading to $\boldsymbol{o}_t^{i_t\mathcal{C}} = [\boldsymbol{o}_t^{i_t}, \boldsymbol{a}_t]^\mathsf{T}$ for $t = \{1, \dots, T-1\}$, and $\boldsymbol{o}_T^{i_t\mathcal{C}} = \boldsymbol{o}_T^{i_t}$ for the last time-step $T$. Analogous to standard latent-variable models, our goal is to learn latent transitions that maximize the marginal likelihood of observations as $\max \log p(\boldsymbol{o}_1^{i_t\mathcal{C}}, \dots, \boldsymbol{o}_T^{i_t\mathcal{C}}) = \max \log p(\boldsymbol{o}_1^{i_t}, \boldsymbol{a}_1, \dots, \boldsymbol{o}_{T-1}^{i_t}, \boldsymbol{a}_{T-1}, \boldsymbol{o}_T^{i_t})$. According to the graphical model in Fig. (1), observations

are generated from latent variables which are temporally evolving. Hence, we regard observations as a resultant of a process in which latent states have been marginalized-out:

$$p\left(\boldsymbol{o}_{1:T}^{\mathcal{MC}}\right) = \prod_{i_t=1}^{N} p\left(\boldsymbol{o}^{i_t \mathcal{C}}\right) = \prod_{i_t=1}^{N} \int_{\boldsymbol{s}_1} \dots \int_{\boldsymbol{s}_T} p(\boldsymbol{o}_1^{i_t}, \boldsymbol{a}_1, \dots, \boldsymbol{o}_{T-1}^{i_t}, \boldsymbol{a}_{T-1}, \boldsymbol{o}_T^{i_t}, \underbrace{\boldsymbol{s}_1, \dots, \boldsymbol{s}_T}_{\text{latent variables}}) \underbrace{d\boldsymbol{s}_1 \dots d\boldsymbol{s}_T}_{\text{marginalization}},$$

where $\boldsymbol{o}_{1:T}^{\mathcal{MC}}$ collects all multi-view observations and actions across time-steps. To devise an algorithm that reasons about latent dynamics, two components need to be better understood. The first relates to factoring our joint density, while the second to approximating multi-dimensional integrals. To factorize our joint density, we follow the modeling assumptions in Fig. (1) and write:

$$p\left(\boldsymbol{o}_{1:T}^{\mathcal{MC}}, \boldsymbol{s}_1, \dots, \boldsymbol{s}_T\right) = \prod_{i_t=1}^{N} p_{\boldsymbol{\theta}_1^{i_t}}(\boldsymbol{s}_1) \prod_{t=2}^{T} p_{\boldsymbol{\theta}_2^{i_t}}(\boldsymbol{o}_t^{i_t}|\boldsymbol{s}_t) p_{\boldsymbol{\theta}_3^{i_t}}(\boldsymbol{s}_t|\hat{\mathcal{H}}_t),$$

where in the last step we introduced $\boldsymbol{\theta}^{i_t} = \{\boldsymbol{\theta}_1^{i_t}, \boldsymbol{\theta}_2^{i_t}, \boldsymbol{\theta}_3^{i_t}\}$ to emphasize modeling parameters that need to be learnt, and $\hat{\mathcal{H}}_t$ to concatenate state and action histories back to $\boldsymbol{s}_1$.

Having dealt with density factorization, another problem to circumvent is that of computing intractable multi-dimensional integrals. This can be achieved by introducing a variational distribution over latent variables, $q_{\boldsymbol{\phi}}(\boldsymbol{s}_1, \dots, \boldsymbol{s}_T|\boldsymbol{o}_{1:T}^{\mathcal{MC}})$, which transform integration into an optimization problem as

$$\log p\left(\boldsymbol{o}_{1:T}^{\mathcal{MC}}\right) = \log \int_{\boldsymbol{s}_{1:T}} \frac{q_{\boldsymbol{\phi}}(\boldsymbol{s}_1, \dots, \boldsymbol{s}_T|\boldsymbol{o}_{1:T}^{\mathcal{MC}})}{q_{\boldsymbol{\phi}}(\boldsymbol{s}_1, \dots, \boldsymbol{s}_T|\boldsymbol{o}_{1:T}^{\mathcal{MC}})} p(\boldsymbol{o}_{1:T}^{\mathcal{MC}}, \boldsymbol{s}_1, \dots, \boldsymbol{s}_T) d\boldsymbol{s}_{1:T}$$

$$\geq \int_{\boldsymbol{s}_{1:T}} q_{\boldsymbol{\phi}}(\boldsymbol{s}_1, \dots, \boldsymbol{s}_T|\boldsymbol{o}_{1:T}^{\mathcal{MC}}) \log \left[\frac{p(\boldsymbol{o}_{1:T}^{\mathcal{MC}}, \boldsymbol{s}_1, \dots, \boldsymbol{s}_T)}{q_{\boldsymbol{\phi}}(\boldsymbol{s}_1, \dots, \boldsymbol{s}_T|\boldsymbol{o}_{1:T}^{\mathcal{MC}})}\right] d\boldsymbol{s}_{1:T},$$

where we used the concavity of the logarithm and Jensen's inequality in the second step of the derivation. We assume a mean-field decomposition for the variational distribution, $q_{\boldsymbol{\phi}}(\boldsymbol{s}_1, \dots, \boldsymbol{s}_T|\boldsymbol{o}_{1:T}^{\mathcal{MC}}) = \prod_{i_t=1}^{N} q_{\boldsymbol{\phi}_1^{i_t}}(\boldsymbol{s}_1) \prod_{t=2}^{T} q_{\boldsymbol{\phi}_t^{i_t}}(\boldsymbol{s}_t|\mathcal{H}_t)$, with $\mathcal{H}_t$ being the observation and action history. This leads to:

$$\log p\left(\boldsymbol{o}_{1:T}^{\mathcal{MC}}\right) \geq \sum_{i_t=1}^{N} \sum_{t=1}^{T} \left[\mathbb{E}_{q_{\boldsymbol{\phi}_t^{i_t}}}\left[\log p_{\boldsymbol{\theta}_2^{i_t}}(\boldsymbol{o}_t^{\mathcal{M}}|\boldsymbol{s}_t)\right] - \text{KL}\left(q_{\boldsymbol{\phi}_t^{i_t}}(\boldsymbol{s}_t|\mathcal{H}_t)||p_{\boldsymbol{\theta}_3^{i_t}}(\boldsymbol{s}_t|\hat{\mathcal{H}}_t)\right)\right]$$

$$- \text{KL}\left(q_{\boldsymbol{\phi}_1^{i_t}}(\boldsymbol{s}_1)||p_{\boldsymbol{\theta}_1^{i_t}}(\boldsymbol{s}_1)\right),$$

where $\text{KL}(p||q)$ denotes the Kullback–Leibler divergence between two distribution. Assuming shared variational parameters (e.g., one variational network), model learning can be formulated as:

$$\max_{\boldsymbol{\theta}_{\text{m}}, \boldsymbol{\phi}} \sum_{i_t=1}^{N} \sum_{t=1}^{T} \left[\mathbb{E}_{q_{\boldsymbol{\phi}^{i_t}}(\boldsymbol{s}_t|\mathcal{H}_t)}\left[\log p_{\boldsymbol{\theta}^{i_t}}(\boldsymbol{o}_t^{i_t}|\boldsymbol{s}_t)\right] - \text{KL}\left(q_{\boldsymbol{\phi}^{i_t}}(\boldsymbol{s}_t|\mathcal{H}_t)||p_{\boldsymbol{\theta}^{i_t}}(\boldsymbol{s}_t|\hat{\mathcal{H}}_t)\right)\right]. \quad (3)$$

Intuitively, Eq. (3) fits the model by maximizing multi-view observation likelihood, while being regularized through the KL-term. Clearly, this is similar to the standard evidence lower-bound with additional components related to handling multi-view types of observations.

### 3.2.2 Distribution Parameterization and Implementation Details

To finalize our problem definition, choices for the modeling and variational distributions can ultimately be problem-dependent. To encode transitions beyond Markov assumptions, we use a memory-based model $g_{\boldsymbol{\psi}}$ (e.g., a recurrent neural network) to serve as the history encoder and future predictor, i.e., $\boldsymbol{h}_t = g_{\boldsymbol{\psi}}(\boldsymbol{s}_{t-1}, \boldsymbol{h}_{t-1}, \boldsymbol{a}_{t-1})$. Introducing memory splits the model into stochastic and deterministic parts, where the deterministic part is the memory model $g_{\boldsymbol{\psi}}$, while the stochastic part is the conditional prior distribution on latent states $\boldsymbol{s}_t$, i.e., $p_{\boldsymbol{\theta}^{i_t}}(\boldsymbol{s}_t|\boldsymbol{h}_t)$. We assume that this distribution is Gaussian with its mean and variance parameterized by a feed-forward neural network taking $\boldsymbol{h}_t$ as inputs.

As for the observation model, the exact form is domain-specific depending on available observation types. In the case when our observation is a low-dimensional vector, we chose a Gaussian parameterization with mean and variance output by a feed-forward network as above. When dealing with images, we parameterized the mean by a deconvolutional neural network [13] and kept an identity covariance. The variational distribution $q_{\boldsymbol{\phi}}$ can thus, respectively, be parameterized by a feed-forward neural network and a convolutional neural network [18] for these two types of views.

With the above assumptions, we can now derive the training loss used in our experiments. First, we rewrite Eq. (3) as

$$\max_{\boldsymbol{\theta}_{\text{m}}, \boldsymbol{\phi}} \sum_{i_t=1}^{N} \sum_{t=1}^{T} \left[\mathbb{E}_{q^{i_t}}\left[\log p_{\boldsymbol{\theta}^{i_t}}(\boldsymbol{o}_t^{i_t}|\boldsymbol{s}_t)\right] + \mathbb{E}_{q^{i_t}}\left[\log p_{\boldsymbol{\theta}^{i_t}}(\boldsymbol{s}_t|\boldsymbol{h}_t)\right] + \mathbb{H}\left[q^{i_t}\right]\right], \quad (4)$$

where $\mathbb{H}$ denotes the entropy, and $q^{i_t}$ represents $q_{\boldsymbol{\phi}^{i_t}}(\boldsymbol{s}_t|\boldsymbol{h}_t,\boldsymbol{o}_t)$. From the first two terms in Eq. (4), we realize that the optimization problem for multi-view model learning consists of two parts: 1) observation reconstruction, 2) transition prediction. Observation reconstruction operates by: 1) inferring the latent state $\boldsymbol{s}_t$ from the observation $\boldsymbol{o}_t$ using the variational model, and 2) decoding $\boldsymbol{s}_t$ to $\tilde{\boldsymbol{o}}_t$ (an approximation of $\boldsymbol{o}_t$). Transition predictions, on the other hand, operate by feeding the previous latent state $\boldsymbol{s}_{t-1}$ and the previous action $\boldsymbol{a}_{t-1}$ to predict the next latent state $\hat{\boldsymbol{s}}_t$ via the memory model. Both parts are optimized by maximizing the log-likelihood under a Gaussian distribution with unit variance. This equates to minimizing the mean squared error between model outputs and actual variable value:

$$\mathcal{L}_r(\boldsymbol{\theta}^{i_t},\boldsymbol{\phi}^{i_t}) = \sum_{i_t=1}^{N}\sum_{t=1}^{T} -\mathbb{E}_{q^{i_t}}\left[\log p_{\boldsymbol{\theta}^{i_t}}(\boldsymbol{o}_t^{i_t}|\boldsymbol{s}_t)\right] = \sum_{i_t=1}^{N}\sum_{t=1}^{T} \|\tilde{\boldsymbol{o}}_t^{i_t} - \boldsymbol{o}_t^{i_t}\|_2,$$

$$\mathcal{L}_p(\boldsymbol{\psi},\boldsymbol{\theta}^{i_t},\boldsymbol{\phi}^{i_t}) = \sum_{i_t=1}^{N}\sum_{t=1}^{T} -\mathbb{E}_{q^{i_t}}\left[\log p_{\boldsymbol{\theta}^{i_t}}(\boldsymbol{s}_t|\boldsymbol{h}_t)\right] = \sum_{i_t=1}^{N}\sum_{t=1}^{T} \|\hat{\boldsymbol{s}}_t^{i_t} - \boldsymbol{s}_t^{i_t}\|_2,$$

where $\|\cdot\|_2$ is the Euclidean norm.

Optimizing Eq. (4) also requires maximizing the entropy of the variational model $\mathbb{H}[q_{\boldsymbol{\phi}^{i_t}}(\boldsymbol{s}_t|\boldsymbol{h}_t,\boldsymbol{o}_t)]$. Intuitively, the variational model aims to increase the element-wise similarity of the latent state $\boldsymbol{s}$ among corresponding observations [14]. Thus, we represent the entropy term as:

$$\mathcal{L}_{\mathbb{H}}(\boldsymbol{\theta}^{i_t},\boldsymbol{\phi}^{i_t}) = \sum_{i_t=1}^{N}\sum_{t=1}^{T} -\mathbb{H}\left[q_{\boldsymbol{\phi}^{i_t}}(\boldsymbol{s}_t|\boldsymbol{h}_t,\boldsymbol{o}_t)\right] \equiv \sum_{i_t=2}^{N}\sum_{t=1}^{T} \|\bar{\boldsymbol{\mu}}_t^{i_t} - \bar{\boldsymbol{\mu}}_t^{1}\|_2, \tag{5}$$

where $\bar{\boldsymbol{\mu}}^{i_t} \in \mathbb{R}^K$ is the average value of the mean of the diagonal Gaussian representing $q_{\boldsymbol{\phi}^{i_t}}(\boldsymbol{s}_t|\boldsymbol{h}_t,\boldsymbol{o}_t)$ for each training batch.

### 3.2.3 Policy Transfer and Few-Shot Reinforcement Learning

As introduced in Section 2.2, trajectory densities in MVRL generalize to support multiple state views by requiring different observation models through time. Such a generalization enables us to achieve cross-view policy transfer and few-shot RL, where we only require very few data from a specific view to train the multi-view model. This can then be used for action selection by: 1) inferring the corresponding latent state, and 2) feeding the latent state into the policy learned from another view with greater accessibility. Details can be found in Appendix A.

Concretely, our learned models $\boldsymbol{\theta}^{i_t}$ should be able to reconstruct corresponding observations for views with shared underlying dynamics (latent state $\boldsymbol{s}$). During model learning, we thus validate the variational and observation model by: 1) inferring the latent state $\boldsymbol{s}$ from the first view's observation $\boldsymbol{o}^1$, and 2) comparing the reconstructed corresponding observation from other views $\tilde{\boldsymbol{o}}^{i_t}$ with the actual observation $\boldsymbol{o}^{i_t}$ through calculating the transformation loss: $\mathcal{L}_t = \sum_{i_t=2}^{N}\|\tilde{\boldsymbol{o}}^{i_t} - \boldsymbol{o}^{i_t}\|_2$. Similarly, the memory model can be validated by: 1) reconstructing the predicted latent state $\hat{\boldsymbol{s}}^1$ of the first view using the observation model of other views to get $\hat{\boldsymbol{o}}^{i_t}$, and 2) comparing $\hat{\boldsymbol{o}}^{i_t}$ with the actual observation $\boldsymbol{o}^{i_t}$, through calculating prediction transformation losses: $\mathcal{L}_{pt} = \sum_{i_t=2}^{N}\|\hat{\boldsymbol{o}}^{i_t} - \boldsymbol{o}^{i_t}\|_2$.

## 4 Experiments

We evaluate our method on a variety of dynamical systems varying in dimensions of their state representation. We consider both high and low dimensional problems to demonstrate the effectiveness of our model-free and model-based solutions. On the model-free side, we demonstrate performance against state-of-the-art methods, such as Proximal Policy Optimization (PPO) [29]. On the model-based side, we are interested in knowing whether our model successfully learns shared dynamics across varying views, and if these can then be utilized to enable efficient control.

We consider dynamical systems from the Atari suite, Roboschool [29], PyBullet [8], and the Highway environments [20]. We generate varying views either by transforming state representations, introducing noise, or by augmenting state variables with additional dummy components. When considering the game of Pong, we allow for varying observations by introducing various transformations to the image representing the state, e.g., rotation, flipping, and horizontal swapping. We then compare our multi-view model with a state-of-the-art modeling approach titled World Models [16]; see Section 4.1. Given successful modeling results, we commence to demonstrate control in both mode-free and model-based scenarios in Section 4.2. Our results demonstrate that although multi-view model-free algorithms can present advantages when compared to standard RL, multi-view model-based techniques are highly more efficient in terms of sample complexities.

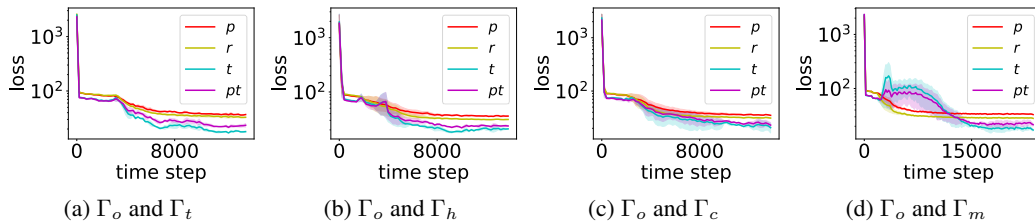

(a) $\Gamma_o$ and $\Gamma_t$      (b) $\Gamma_o$ and $\Gamma_h$      (c) $\Gamma_o$ and $\Gamma_c$      (d) $\Gamma_o$ and $\Gamma_m$

Figure 2: Training multi-view models on Atari Pong. Legend: $p$: prediction loss $\mathcal{L}_p$; $r$: reconstruction loss $\mathcal{L}_r$; $t$: transformation loss $\mathcal{L}_t$; $pt$: predicted transformation loss $\mathcal{L}_{pt}$. These results demonstrate that our method correctly converges in terms of loss values.

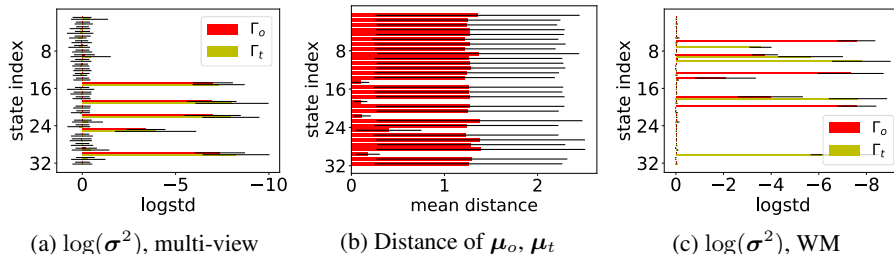

(a) $\log(\boldsymbol{\sigma}^2)$, multi-view      (b) Distance of $\boldsymbol{\mu}_o, \boldsymbol{\mu}_t$      (c) $\log(\boldsymbol{\sigma}^2)$, WM

Figure 3: Difference between inferred latent states from $\Gamma_o$ and $\Gamma_t$. Results demonstrating that our method is capable of learning key elements – a property essential for multi-view dynamics learning. These results also demonstrate that extracting such key-elements is challenging for world-models.

## 4.1 Modeling Results

To evaluate multi-view model learning we generated five views by varying state representations (i.e., images) in the Atari Pong environment. We kept dynamics unchanged and considered four sensory transformations of the observation frame $\Gamma_o$. Namely, we considered varying views as: 1) transposed images $\Gamma_t$, 2) horizontally-swapped images $\Gamma_h$, 3) inverse images $\Gamma_c$, and 4) mirror-symmetric images $\Gamma_m$. Exact details on how these views have been generated can be found in Appendix C.1.1.

For simplicity, we built pair-wise multi-view models between $\Gamma_o$ and one of the above five variants. Fig. (2) illustrates convergence results of multi-view prediction and transformation for different views of Atari Pong. Fig (3) further investigates the learnt shared dynamics among different views ($\Gamma_o$ and $\Gamma_t$). Fig. (3a) illustrates the converged $\log(\boldsymbol{\sigma}^2)$, and the log standard deviation of the latent state variable, in $\Gamma_o$ and $\Gamma_t$. Observe that a small group of elements (indexed as 14, 18, 21, 24 and 29) have relatively low variance in both views, thus keeping stable values across different observations.

We consider these elements as the critical part in representing the shared dynamics and define them as *key elements*. Clearly, learning a shared group of *key elements* across different views is the target in the multi-view model. Results in Fig. (3b), illustrate the distance between $\boldsymbol{\mu}_o$ and $\boldsymbol{\mu}_t$ for the multi-view model demonstrating convergence. As the same group of elements are, in fact, close to *key elements* learnt by the multi-view model, we conclude that we can capture shared dynamics across different views. Further analysis of these key elements is also presented in the appendix.

In Fig. (3c), we also report the converged value of $\log(\boldsymbol{\sigma}^2)$ of World Models (WM) [16] under the multi-view setting. Although the world model can still infer the latent dynamics of both environments, the large difference between learnt dynamics demonstrates that varying views resemble a challenge to world models – our algorithm, however, is capable of capturing such hidden shared dynamics.

## 4.2 Policy Learning Results

Given successful modeling results, in this section we present results on controlling systems across multiple views within a RL setting. Namely, we evaluate our multi-view RL approach on several high and low dimensional tasks. Our systems consisted of: 1) Cartpole ($\mathcal{O} \subseteq \mathbb{R}^4, \boldsymbol{a} \in \{0,1\}$), where the goal is to balance a pole by applying left or right forces to a pivot, 2) hopper ($\mathcal{O} \subseteq \mathbb{R}^{15}, \mathcal{A} \subseteq \mathbb{R}^3$), where the focus is on locomotion such that the dynamical system hops forward as fast as possible, 3) RACECAR ($\mathcal{O} \subseteq \mathbb{R}^2, \mathcal{A} \subseteq \mathbb{R}^2$), where the observation is the position (x,y) of a randomly placed ball in the camera frame and the reward is based on the distance to the ball,

and 4) parking ($\mathcal{O} \subseteq \mathbb{R}^6, \mathcal{A} \subseteq \mathbb{R}^2$), where an ego-vehicle must park in a given space with an appropriate heading (a goal-conditioned continuous control task). The evaluation metric we used was defined as the average testing return) across all views with respect to the amount of training samples (number of interactions). We use the same setting in all experiments to generate multiple views. Namely, the original environment observation is used as the first view,

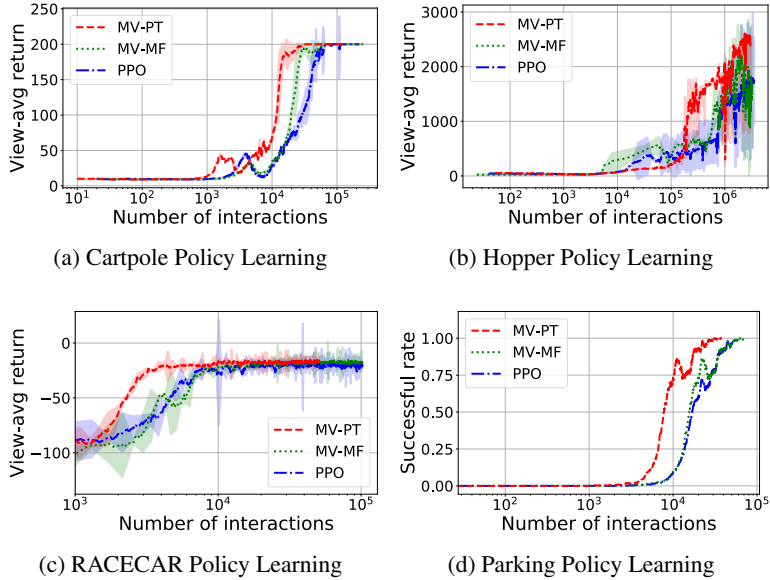

(a) Cartpole Policy Learning  (b) Hopper Policy Learning

(c) RACECAR Policy Learning  (d) Parking Policy Learning

Figure 4: Policy learning results demonstrating that our method outperforms others in terms of sample complexities.

and adding dummy dimensions (two dims) and large-scale noises (0.1 after observation normalization) to the original observation generates the second view. Such a setting would allow us to understand if our model can learn shared dynamics with mis-specified state representations, and if such a model can then be used for control.

For all experiments, we trained the multi-view model with a few samples gathered from all views and used the resultant for policy transfer (MV-PT) between views during the test period. We chose state-of-the-art PPO [29] – an algorithm based on the work in [27], as the baseline by training separate models on different views and aggregated results together. The multi-view model-free (MV-MF) method is trained by augmenting PPO with concatenated observations. Relevant parameter values and implementation details are listed in the Appendix C.2.

Fig. (4) shows the result of average testing return (the average testing successful rate for the parking task) from all views. On the Cartpole and parking tasks, our MV-MF algorithms can present improvements when compared to strong model-free baselines such as PPO, showing the advantage of leveraging information from multiple views than training independently within each view. On the other hand, multi-view model-based techniques give the best performance on all tasks and reduce number of samples needed by around two orders of magnitudes in most tasks. This proves that MV-PT greatly reduces the required amount of training samples to reach good performance.

We also conducted pure model-based RL experiment and compared our multi-view dynamic model against 1) PILCO [9], which can be regarded as state-of-the-art model-based solution using a Gaussian Process

Table 1: Model-based RL result on the Cartpole.

| PILCO | PILCO-aug | MLP | MV-MB |
|-------|-----------|-----|-------|
| 240 | 320 | $2334 \pm 358$ | $381 \pm 28$ |

dynamic model, 2) PILCO with augmented states, i.e., a single PILCO model to approximate the data distribution from all views, and 3) a multilayer perceptron (MLP). We use the same planning algorithm for all model-based methods, e.g., hill climbing or Model Predictive Control (MPC) [25], depending on the task at hand. Table 1 shows the result on the Cartpole environment, where we evaluate all methods by the amount of interactions till success. Each training rollout has at most 40 steps of interactions and we define the success as reaching an average testing return of 195.0. Although multi-view model performs slightly worse than PILCO in the Cartpole task, we found out that model-based alone cannot perform well on tasks without suitable environment rewards. For example, the reward function in hopper primarily encourages higher speed and lower energy consumption. Such high-level reward functions make it hard for model-based methods to succeed; therefore, the results of model-based algorithms on all tasks are lower than using other specifically designed reward functions. Tailoring reward functions, though interesting, is out of the scope of this paper. MV-PT, on the other hand, outperforms others significantly, see Fig. (4).

# 5 Related Work

Our work has extended model-based RL to the multi-view scenario. Model-based RL for POMDPs have been shown to be more effective than model-free alternatives in certain tasks [37, 35, 21, 19]. One of the classical combination of model-based and model-free algorithms is Dyna-Q [31], which learns the policy from both the model and the environment by supplementing real world on-policy experiences with simulated trajectories. However, using trajectories from a non-optimal or biased model can lead to learning a poor policy [15]. To model the world environment of Atari Games, autoencoders have been used to predict the next observation and environment rewards [19]. Some previous works [28, 16, 19] maintain a recurrent architecture to model the world using unsupervised learning and proved its efficiency in helping RL agents in complex environments. Mb-Mf [25] is a framework bridging the gap between model-free and model-based methods by employing MPC to pre-train a policy within the learned model before training it with standard model-free method. However, these models can only be applied to a single environment and need to be built from scratch for new environments. Although using a similar recurrent architecture, our work differs from above works by learning the shared dynamics over multiple views. Also, many of the above advancements are orthogonal to our proposed approach, which can definitely benefit from model ensemble, e.g., pre-train the model-free policy within the multi-view model when reward models are accessible.

Another related research area is multi-task learning (or meta-learning). To achieve multi-task learning, recurrent architectures [10, 36] have also been used to learn to reinforcement learn by adapting to different MDPs automatically. These have been shown to be comparable to the UCB1 algorithm [4] on bandit problems. Meta-learning shared hierarchies (MLSH) [12] share sub-policies among different tasks to achieve the goal in the training process, where high hierarchy actions are obtained and reused in other tasks. Model-agnostic meta-learning algorithm (MAML) [11] minimizes the total error across multiple tasks by locally conducting few-shot learning to find the optimal parameters for both supervised learning and RL. Actor-mimic [26] distills multiple pre-trained DQNs on different tasks into one single network to accelerate the learning process by initializing the learning model with learned parameters of the distilled network. To achieve promising results, these pre-trained DQNs have to be expert policies. Distral [32] learns multiple tasks jointly and trains a shared policy as the "centroid" by distillation. Concurrently with our work, ADRL [5] has extended model-free RL to multi-view environments and proposed an attention-based policy aggregation method based on the Q-value of the actor-critic worker for each view. Most of above approaches consider the problems within the model-free RL paradigm and focus on finding the common structure in the policy space. However, model-free approaches require large amounts of data to explore in high-dimensional environments. In contrast, we explicitly maintain a multi-view dynamic model to capture the latent structures and dynamics of the environment, thus having more stable correlation signals.

Some algorithms from meta-learning have been adapted to the model-based setting [1, 7]. These focused on model adaptation when the model is incomplete, or the underlying MDPs are evolving. By taking the unlearnt model as a new task and continuously learning new structures, the agent can keep its model up to date. Different from these approaches, we focus on how to establish the common dynamics over compact representations of observations generated from different emission models.

# 6 Conclusions

In this paper, we proposed multi-view reinforcement learning as a generalization of partially observable Markov decision processes that exhibit multiple observation densities. We derive model-free and model-based solutions to multi-view reinforcement learning, and demonstrate the effectiveness of our method on a variety of control benchmarks. Notably, we show that model-free multi-view methods through observation augmentation significantly reduce number of training samples when compared to state-of-the-art reinforcement learning techniques, e.g., PPO, and demonstrate that model-based approaches through cross-view policy transfer allow for extremely efficient learners needing significantly fewer number of training samples.

There are multiple interesting avenues for future work. First, we would like to apply our technique to real-world robotic systems such as self-driving cars, and second, use our method for transferring between varying views across domains.

## Footnotes

†Honorary Lecturer at University College London

[3]We note that other titles for this work are also possible, e.g., Multi-View POMDPs, or Multi-Observation POMDPs. The emphasis is on the fact that little literature has considered RL with varying sensory observations.

[4]It is worth noting that our experiments reveal that PILCO, for instance, faces challenges when dealing with high-dimensional systems, e.g., Hopper. In future, we plan to investigate latent space Gaussian process models with the aim of carrying sample efficient algorithms to high-dimensional systems.

[5]Please note it is possible to incorporate the actions in $\mathcal{H}_t$ by simply introducing additional action variables.

[6]Different from the model-free solver introduced in Sect. 3.1, we don't assume the accessibility to all views.

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
