[Supplementary Material · nips_lmn_supp.pdf]

# A Model-based Multi-view Reinforcement Learning Algorithm

For completeness, we provide the Model-based Multi-view reinforcement learning algorithm below.

---

**Algorithm 1** Model-based Multi-view Reinforcement Learning through Model Predictive Control

---
1: gather dataset $\mathcal{D}_{\text{RAND}}$ of random trajectories
2: initialize empty dataset $\mathcal{D}_{\text{RL}}$, MPC planning horizon $H$, and randomly initialize $\psi, \boldsymbol{\theta}^{i_t}, \boldsymbol{\phi}^{i_t}$ for $i_t \in \{1, \dots, N\}$
3: **for** iter=1 **to** max_iter **do**
4:     train $\psi, \boldsymbol{\theta}^{i_t}, \boldsymbol{\phi}^{i_t}$ by performing gradient descent on $\mathcal{L}_r(\boldsymbol{\theta}^{i_t}, \boldsymbol{\phi}^{i_t}), \mathcal{L}_p(\psi, \boldsymbol{\theta}^{i_t}, \boldsymbol{\phi}^{i_t})$, and $\mathcal{L}_{\mathbb{H}}(\boldsymbol{\theta}^{i_t}, \boldsymbol{\phi}^{i_t})$, using $\mathcal{D}_{\text{RAND}}$ and $\mathcal{D}_{\text{RL}}$
5:     **for** $t = 1$ **to** $T$ **do**
6:         get agent's current observation $\boldsymbol{o}_t^{i_t}$ from an available view $i_t$
7:         infer agent's current state $\boldsymbol{s}_t$ using $\boldsymbol{\phi}^{i_t}$
8:         use $\psi, \boldsymbol{\theta}^{i_t}, \boldsymbol{\phi}^{i_t}$ to estimate optimal action sequence

$$\mathbf{A}_t^{(H)} = \arg\max_{\mathbf{A}_t^{(H)}} \sum_{t'=t}^{t+H-1} r(\hat{\boldsymbol{o}}_{t'}^{i_{t'}}, \boldsymbol{a}_{t'}),$$

    where $\hat{\boldsymbol{o}}_t^{i_t} = \boldsymbol{o}_t^{i_t}, \hat{\boldsymbol{o}}_{t'+1}^{i_{t'+1}} \sim p_{\boldsymbol{\theta}^{i_{t'+1}}}(\boldsymbol{o}_{t'+1}|\boldsymbol{h}_{t'+1}), \boldsymbol{h}_{t'+1} = g_{\psi}(\boldsymbol{s}_{t'}, \boldsymbol{h}_{t'}, \boldsymbol{a}_{t'}), \boldsymbol{s}_{t'} \sim q_{\boldsymbol{\phi}^{i_{t'}}}(\boldsymbol{s}_{t'}|\boldsymbol{h}_{t'})$
9:         execute first action $\boldsymbol{a}_t$ from selected action sequence $\mathbf{A}_t^{(H)}$
10:        add $(\boldsymbol{o}_t^{i_t}, \boldsymbol{a}_t)$ to $\mathcal{D}_{\text{RL}}$
11:     **end for**
12: **end for**

---

---

**Algorithm 2** Model-based Multi-view Reinforcement Learning through Policy Transfer (MV-PT)

---
1: gather dataset $\mathcal{D}_{\text{RAND}}$ of random trajectories
2: initialize empty dataset $\mathcal{D}_{\text{RL}}$, model-free policy $\boldsymbol{\omega}$, and randomly initialize $\psi, \boldsymbol{\theta}^{i_t}, \boldsymbol{\phi}^{i_t}$ for $i_t \in \{1, \dots, N\}$
3: **for** iter=1 **to** max_iter **do**
4:     train $\psi, \boldsymbol{\theta}^{i_t}, \boldsymbol{\phi}^{i_t}$ by performing gradient descent on $\mathcal{L}_r(\boldsymbol{\theta}^{i_t}, \boldsymbol{\phi}^{i_t}), \mathcal{L}_p(\psi, \boldsymbol{\theta}^{i_t}, \boldsymbol{\phi}^{i_t})$, and $\mathcal{L}_{\mathbb{H}}(\boldsymbol{\theta}^{i_t}, \boldsymbol{\phi}^{i_t})$, using $\mathcal{D}_{\text{RAND}}$ and $\mathcal{D}_{\text{RL}}$ for target view(s) and policy learning view(s)
5:     **for** $t = 1$ **to** $T$ **do**
6:         get agent's current observation $\boldsymbol{o}_t^{i_t}$ from an available view $i_t$
7:         infer agent's current state $\boldsymbol{s}_t$ using $\boldsymbol{\phi}^{i_t}$
8:         train $\boldsymbol{\omega}$ by performing gradient descent based on a standard model-free algorithm using $\boldsymbol{s}_t$ and $\boldsymbol{a}_t$
9:         add $(\boldsymbol{o}_t^{i_t}, \boldsymbol{a}_t)$ to $\mathcal{D}_{\text{RL}}$
10:        **if** need to act in view $i_t$ **then**
11:            get action $\boldsymbol{a}_t$ from $\boldsymbol{\omega}$ using $\boldsymbol{s}_t$
12:        **end if**
13:     **end for**
14: **end for**

---

# B Derivatives of Multi-view Model-free Policy Gradient

The type of algorithm we employ for model-free multi-view reinforcement learning falls in the class of policy gradient algorithms, which update the agent's policy-defining parameters $\boldsymbol{\omega} \in \mathbb{R}^d$ directly by estimating a gradient in the direction of higher reward. Given the problem definition in Equation 1, the gradient of the loss with respect to the network parameters $\boldsymbol{\omega}$ can be computed as:

$$\nabla_{\boldsymbol{\omega}} \mathbb{E}_{\boldsymbol{\tau}^{\mathcal{M}}} \left[ \mathcal{R}_T \left( \boldsymbol{\tau}^{\mathcal{M}} \right) \right] = \int p_{\pi_{\boldsymbol{\omega}}^{\mathcal{M}}} \left( \boldsymbol{\tau}^{\mathcal{M}} \right) \nabla_{\boldsymbol{\omega}} \log \left[ p_{\pi_{\boldsymbol{\omega}}^{\mathcal{M}}} \left( \boldsymbol{\tau}^{\mathcal{M}} \right) \right] \mathcal{R}_T \left( \boldsymbol{\tau}^{\mathcal{M}} \right) d\boldsymbol{\tau}^{\mathcal{M}}$$

$$= \mathbb{E}_{\boldsymbol{\tau}^{\mathcal{M}}} \left[ \nabla_{\boldsymbol{\omega}} \log \left[ p_{\pi_{\boldsymbol{\omega}}^{\mathcal{M}}} \left( \boldsymbol{\tau}^{\mathcal{M}} \right) \right] \mathcal{R}_T \left( \boldsymbol{\tau}^{\mathcal{M}} \right) \right],$$

(a) $\Gamma_o \to \Gamma_t$      (b) $\Gamma_o \to \Gamma_h$      (c) $\Gamma_o \to \Gamma_c$      (d) $\Gamma_o \to \Gamma_m$

Figure 5: Atari Pong variants.

where we used the "likelihood-ratio" trick in the third step of the derivation. Now, one can proceed by taking a sample average of the gradient using Monte Carlo to update the policy parameters $\boldsymbol{\omega}$, suggesting the following update rule:

$$\boldsymbol{\omega}^{k+1} \approx \boldsymbol{\omega}^k + \eta^k \frac{1}{M} \sum_{j=1}^{M} \nabla_{\boldsymbol{\omega}} \log \left[ p_{\pi_{\boldsymbol{\omega}}^{\mathcal{M}}} \left( \boldsymbol{\tau}_j^{\mathcal{M}} \right) \right] \mathcal{R}_T \left( \boldsymbol{\tau}_j^{\mathcal{M}} \right).$$

Mote Carlo estimation above is a fast approximation of the gradient for the current policy with convergence speed of $\mathcal{O} \left( \frac{1}{\sqrt{M}} \right)$ to the true gradient independent of the number of parameters of the policy. It is also worth noting that although the trajectory distribution depends on the unknown initial state distribution, unknown observation models, and hidden state dynamics, the gradient only includes policy components that can be controlled by the agent.

Though fast in convergence to the true gradient, Monte Carlo estimates suffer from high variance, e.g., it is easy to show that variance grows linearly in the time horizon. Unfortunately, the naive approach of sampling big-enough batch sizes is not an option in reinforcement learning due to the high cost of collecting samples, i.e., interacting with the environment. For this reason, literature has focused on introducing baselines aiming to reduce variance [46, 48]. We follow a similar approach here, and introduce an observation based baseline to reduce the variance of our gradient estimate. Our baseline, $\mathcal{B}_\phi(\mathcal{H}_t)$, will take as inputs observations and actions[7], and learn to predict future returns given the current policy. Such a baseline can easily be represented as an LSTM recurrent neural network as noted in [47]. Consequently, we can rewrite our update rule as:

$$\boldsymbol{\omega}^{k+1} = \boldsymbol{\omega}^k + \eta^k \frac{1}{M} \sum_{j=1}^{M} \sum_{t=1}^{T} \nabla_{\boldsymbol{\omega}} \log \pi^{\mathcal{M}} \left( \boldsymbol{a}_t^j | \mathcal{H}_t^j \right) \left( \mathcal{R}(\boldsymbol{s}_t^j, \boldsymbol{a}_t^j) - \mathcal{B}_\phi(\mathcal{H}_t^j) \right). \tag{6}$$

## C    Experiment Details

### C.1    Model learning details

#### C.1.1    View Settings of Atari Pong

As shown in Fig. (5), each variant corresponds to one transformation from $\Gamma_o$: (a) the transposed $\Gamma_t$, which is transformed from the state observation of $\Gamma_o$ by clockwise rotating $90°$ and horizontal flipping; (b) the horizontal-swapped $\Gamma_h$, which is generated by vertically splitting the observation frame of $\Gamma_o$ from the center and swapping the left part with the right part; (c) the inverse $\Gamma_c$, which is created by exchanging the background color with the paddles/ball color of $\Gamma_o$; and (d) the mirror-symmetric $\Gamma_m$, which reflects $\Gamma_o$ like a mirror by horizontally swapping the observation.

#### C.1.2    Multi-view Model Setting

Different from the original Atari Pong observations, we (1) transform each frame to a binary matrix; (2) remove the scoreboard; and (3) resize each frame to $D = 64 * 64$ to serve as the observation of $\Gamma_o$. The action space is formed by all six available discrete actions of the original Atari environment. The observation model adopts the same architecture as a typical VAE with $K = 32$. The memory model is a 32-units LSTM connected to the same output layer of the observation model. We set the batch

Figure 6: Validating the importance of *key elements* in $\Gamma_o$. (a) Sum of absolute weights connected to $s$ in the reconstruction network; (b) Mean absolute gradients of output to $s$ in the reconstruction network.

Table 2: Hyperparameters for PPO

|  | Cartpole | Hopper | RACECAR |
|---|---|---|---|
| Horizon (T) | 2048 | 2048 | 2048 |
| Adam stepsize | $3 * 10^{-4}$ | $3 * 10^{-4}$ | $3 * 10^{-4}$ |
| Num. epochs | 15 | 10 | 15 |
| Mini-batch size | 1024 | 32 | 1024 |
| Discount ($\gamma$) | 0.99 | 0.99 | 0.99 |
| GAE parameter ($\lambda$) | 0.95 | 0.95 | 0.95 |
| Number of actors | 1 | 1 | 8 |
| Clipping parameter $\epsilon$ | $1 * 10^{-5}$ | $1 * 10^{-5}$ | $1 * 10^{-5}$ |
| VF coeff. $c1$ | 0.5 | 0.5 | 0.5 |
| Entropy coeff. $c2$ | 0 | 0 | 0 |

size for each task as 16 and the sequence length of LSTM as 25. We alternate the training process for the multi-view model between minimizing $\mathcal{L}_r$ and $\mathcal{L}_p$ by setting each training iteration with 20 prediction iterations and 10 reconstruction iterations, since the adjustment of the observation model to satisfy the learning of shared dynamics will affect model's reconstruction ability. We explore training the shared dynamics on two views with corresponding inputs and non-corresponding inputs, i.e., using corresponding states from different views as training data or not, to verify the performance of multi-view models.

To collect the training data covering most dynamics of the Pong environment, we use an agent with random policy to play the game for $10,000$ episodes with an episode length of 1000. At each training time step, we randomly sample 16 trajectories of length 25 from the dataset as the training data for $\Gamma_o$, and transform these samples to corresponding observations as the training data for $\Gamma_i$, thus explicitly making the training input for different views share the same transition dynamics.

### C.1.3 Additional Experiment Results

To further validate the importance of key elements in extracting the underlying dynamics, we show the weights of the reconstruction network connected to $s$ of $\Gamma_o$ in Fig. (6a). As the sum of absolute weights connected to key elements are much larger than others, the change of key elements will apply higher influence to the reconstruction output $\tilde{o}$, thus illustrating their significance in latent representations.

We then compute the gradients of the output $\tilde{o}$ with respect to the $s$ to observe which part of $s$ contributes more to the visual stimuli. As shown in Fig. (6b), the mean absolute gradients of key elements are significantly larger, while other elements have nearly zero gradients (no contributions to $\tilde{o}$). Consequently, the shared dynamics are mainly expressed by key elements.

Table 3: Hyperparameters for DDPG and Hindsight Experience Replay

|  | Parking |
| --- | --- |
| Cycles to collect samples | 50 |
| Training batch size | 40 |
| Sample batch size | 256 |
| HER strategy | future |
| Discount ($\gamma$) | 0.98 |
| clip return | 50 |
| actor learning rate | 0.001 |
| critic learning rate | 0.001 |
| average coefficient | 0.95 |
| clip range | 5 |
| num-rollouts-per-mpi | 2 |
| noise $\epsilon$ | 0.2 |
| random $\epsilon$ | 0.3 |
| buffer size | $1 * 10^6$ |
| replay-k | 4 |
| clip-ratio | 200 |

## C.2  Policy learning details

We use the same structure for the multi-view model as mentioned in Sect. C.1. For all environments, we generate 100 initial roll-out trajectories using random policies (except for the Cartpole, where we only generate 20 rollouts). We use PPO as the model-free policy learning algorithm for MV-PT and MV-MF in the Cartpole, hopper and RACECAR tasks, and list the hyperparameters in Table 2. For the parking environment, we use DDPG [45] with hindsight experience replay [43], and list the hyperparameters in Table 3. For model-based baselines, we implement PILCO following the original setting from [44], and choose the MLP with one hidden layer of 128 units and ReLU activation functions.

## Footnotes

[7]Even though this baseline is action-dependent, one can show it to be unbiased. The trick is to realize that a history at time $t$ is independent of action $\boldsymbol{u}_t$ and rather depends on all action up to the $t - 1^{th}$ instance.