[Reviews · NeurIPS 2019]

Reviewer 1



Originality: This work is innovative in generalizing markov decision process to multi-view scenarios. The authors have clearly distinguished their work from the state of the art and similar works. Quality: The submission is technically correct. The claims made in the paper that multi-view RL using policy transfer between views can achieve convergence faster has been adequately demonstrated. The works reads like a complete piece of work and the authors have performed thorough experiments to demonstrate the claims. They did discuss the case where the proposed solution did not better the competing method, providing justification on why that would be the case. Clarity: The paper skips over a lot of derivations and only a person who is very familiar with MVRL would be able to understand it without supplemental document. Otherwise, the paper is well written. However, without the supplementary material, it would be hard to replicate the experiments. Significance: The work seems relevant and significant to advancing the state of the art in reinforcement learning. There are increasing number of scenarios where multiple sensors sense the same environment. Therefore, multi-view RL is definitely going to be a hot topic of research.

Reviewer 2



This work proposes a simple but efficient extension of POMPDs to MV data by extending the observation/state space over the multiple views and modeling the long-term dependencies between the (latent) states. To deal with multi-view latent variables (states), a variational lower bound is derived and optimized in two settings: model-based and model-free (using run/nn-s). The optimization of the derived objective is not trivial and the authors propose various strategies to effectively deal with the dimension of this problem (including the policy transfer and few-shot-learning ideas). I find this work novel and very well presented. The paper is well written and easy to follow. The supplementary materials describe in the algorithm details making it easier to understand the model. The state-of-the-art and related approaches are well reviewed and compared with the proposed. The results on different RL environments show that the proposed can effectively learn underlying shared dynamics (due to the mv modeling), in contrast to the world model.

Reviewer 3



Originality: Considering that multi-view and multi-modal RL papers tend to offer ad-hoc solutions to the problem, this paper's formalization is a nice contribution. Quality & Clarity: The paper is well presented, the math is mostly clear, although some parts aren't obviously translatable to an implementation. In terms of experiments, it seems that many details are lacking, and as far as I call tell, all figures represent a single run of each setting, which is worrisome. Significance: While the contributed framework does seem like a useful formalism, this paper fails to convince me that it actually is: - The proposed experiment in 4.1 creates artificial views which don't seem representative of multimodal settings, in that they all contain the same *information*. It would have been more convincing to feature an experiment where views are truly independent when conditioned on the current state (e.g. dialog and facial expression, partial views). - It's not clear that the advantage comes from your formulation rather than just more things being learned (i.e., what you propose reduces sample complexity because it is an auxiliary task). You should have experiments confirming this. - Again, each experiment setting seems to only have a single run. All your results could be plain luck. Additional comments: - l49, "agents *to* reason" - section 2.1, iiuc, you force upon the agent to only receive information o_t^{i_t} about one view per timestep. What is the distribution of i_t? Is it dependent on state and action? Is it a choice of the agent? This should be clear in your framework. - l94 "existing *advancements* on" - l105 "thus being optimal than independent", what do you mean? "as optimal as"? "more optimal than"? - Figure 2, why are the X axes of different lengths? How did you choose when to stop training? This should be reported - Figure 4, I'm not sure I see the interest of having the X axis be in log-scale. - Table 1, Why the "\sim 360"? Seems like you should be reporting mean and variance, like "360 \pm 10" - l245, what view did you transfer? when? We need more details - l269 "primarily *encourages* higher"

[Author Response · NeurIPS 2019]

We thank all reviewers for their constructive and helpful comments that will allow us to better shape this paper.

# 1   Reviewer I:

We are very thankful for the review and will definitely increase our plot sizes in the final version in case of acceptance.
Also, thank you for pointing out real-world experiments. In fact, we plan to take our approach to robotics in the future.
We believe self-driving cars present an ideal test-bed for our algorithm.

# 2   Reviewer II:

Thank you for the constructive and informative review. We will make sure to increase the figure sizes.

**On Few-Shot Learning**    We achieve few-shot learning through meta-learning. Concretely, our model learns shared
dynamics across different views and could be used as the transition model for learning from a new-coming view.

**On PILCO**    These result are shown on both the Cart-Pole (Table I in the submission) and Hopper (Fig. (4) in the
submission). We also attempted PILCO on Race-Car and the parking environments. Unfortunately, we did not get good
performance on these tasks. We think this is due to the very sparse reward signals in these problems – a setting less
suitable for PILCO and can potentially be remedied by shaping the reward function. We chose to stick standard rewards
and plan to tackle complete model-based RL in the future. We also found that the combination of model-based and
model-free is more suitable for general tasks to reach higher rewards with fewer training data. We will make sure to
clarify these in the main paper as well.

# 3   Reviewer IV:

Thank you for the feedback that will help us improve our paper. We will make sure to fix the typos noted by the
reviewer.

**On the distribution of** $i_t$    We will clarify in the original paper. $i_t$ is allowed to vary from one to N with being the
different types of observations. Each $i_t$ is dependent on the observation function (availability of specific views) and
conditioned on the environment's state.

**On X-Axis and Log-Scale**    We use Log-Scale to compare the performance of model-based (MB) and model-free (MF)
in a single plot (e.g., PILCO and PPO), since MB methods require much fewer samples than MF. We follow a standard
protocol for stopping criteria, where training is halted when the difference of prediction loss $p$ (and reconstruction loss
$r$) between two training steps is less than a threshold (4) for a continuous of 5000 steps. Please note that the Y-Axis of
Fig. (2) is log loss, we will change it to original loss with log-scale.

**On Comparison Against PPO**    We respectfully disagree with the reviewer and firmly believe that our comparison
against PPO is fair. Namely, we compare against PPO by using the same amount of data from the same views. We
showed that our model outperforms PPO in view-average return because it learns the shared latent state and joint
transitions. Auxiliary tasks comparison can be interesting direction in the future but we think current results clearly
demonstrate our method is superior to PPO in a fair manner (both using same amount of data).

**On View Choices for Modeling Experiments**    Our modeling experiments are designed to illustrate the capability
of our model learning the shared latent states and dynamics across different views. We showed that our model could
effectively learn underlying shared dynamics in contrast to the state-of-the-art world model.

**On Toy Problem and More Experiments**    We thank the reviewer for pointing this out. We plan to have a toy problem
to better illustrate the performance. Also, we plan to improve the figure sizes. We also ran multiple runs and have
variance plots. We will add in the main paper. We refrain from showing them here so as to keep with the fairness and
page limit.

[Meta-Review · NeurIPS 2019]

Reviewers are all excited about the extension of model-free and model-based RL to multi-view/multi-modal settings. The authors offer formal formulation of multi-view RL by extending POMDP. The model-free case is rather straight-forward, and the model-based case leads to variational inference. Reviewers however, expressed concern regarding the setup of toy examples and the quality of the evaluation. Overall, the merit of the work outweigh the weakness. Please try to incorporate reviewers' suggestions into your final draft.